# Conserved Sequence Analysis of Influenza A Virus HA Segment and Its Application in Rapid Typing

**DOI:** 10.3390/diagnostics11081328

**Published:** 2021-07-23

**Authors:** Qianyu Lin, Xiang Ji, Feng Wu, Lan Ma

**Affiliations:** 1Tsinghua-Berkeley Shenzhen Institute, Tsinghua University, Shenzhen 518055, China; lingy18@mails.tsinghua.edu.cn; 2Tsinghua Shenzhen International Graduate School, Tsinghua University, Shenzhen 518055, China; hijiri_byakuren@126.com (X.J.); wu8721@126.com (F.W.); 3Shenzhen Bay Laboratory, Shenzhen 518038, China

**Keywords:** influenza 1, multiple sequence alignment 2, conserved sequence 3, rapid detection 4

## Abstract

The high mutation rate of the influenza A virus hemagglutinin segment poses great challenges to its long-term effective testing and subtyping. Our conserved sequence searching method achieves high-specificity conserved sequences on H1–H9 subtypes. In addition, PCR experiments show that primers based on conserved sequences can be used in influenza A virus HA subtyping. Conserved sequence-based primers are expected to be long-term, effective subtyping tools for influenza A virus HA.

## 1. Introduction

Influenza is a common widespread infectious disease that has caused a considerable number of human deaths in the past century [1]. The annual influenza epidemic results in 3 to 5 million cases of severe illness and causes 0.29 to 0.65 million respiratory deaths [2]. Influenza A (referred to as “influenza” in the following text) virus is an RNA virus. The surface antigens of influenza are hemagglutinin (HA) and neuraminidase (NA); 18 different HAs and 11 different NAs are known to date. Influenza’s subtype is defined according to HA and NA subtypes. The influenza virus invades and infects cells via HA’s binding to cells’ sialic acid-containing receptors [3]. Specific binding with hemagglutinin to reduce the infectious capacity of the influenza virus is the main idea of existing treatment and prevention of influenza diseases. Thus, fast and accurate identification of HA subtypes is important for the diagnosis and treatment of influenza.

However, the high mutation rate of influenza virus results in a large number of single-nucleotide polymorphisms (SNPs) and antigenic drift (accumulation) [4,5]. The antigenic drifts result in the antigen’s protein sequence and structure changing, which may reduce the antibody’s specific binding ability with the antigen, finally improving the effectiveness of virus spread among the population vaccinated or immunized to the old strains. Unlike antigenic drift, SNPs do not influence the antibody function, but they can reduce the nucleic acid amplification testing’s specificity and sensitivity for the reason of mutation base mismatch to primers.

Taking conserved sequences as references is also necessary in antibody development for influenza disease treatment. The HA protein, which is one of the main antigens [6] on the surface of the influenza virus, is the key to the virus’s successful binding and its entering host cells. Current antibody development for influenza disease treatment generally focuses on the HA protein; to date, the antibodies targeting the HA stalk [7] and head [8] have allowed considerable progress. However, due to the low degree of sequence conservation of the target binding sites, the existing influenza vaccines and antibodies are sensitive to antigenic drift [9], which causes these vaccines and antibodies to have bad performance in terms of the long-term effect. Of course, the long-term effect of influenza products needs to be robust against mutations, and a conserved sequence is a good choice for both testing primers for nucleic acid amplification and vaccine design [10,11].

Sequence alignment, including pairwise sequence alignment [12,13] and multiple sequence alignment (MSA), is the main method in conserved sequence searching, and it plays an important role in bioinformatics. Different MSA algorithms have different performances in different sequence datasets. Clustal [14] is the most widely used tool for MSA. It calculates the distance matrix by pairwise alignment, builds guide trees, and makes progressive alignment following the guide tree. T-coffee [15] has a higher accuracy result than that of other methods, and it is applicable to small sample sizes and short-length sequence data, but the alignment speed is too slow when used on large-scale datasets. Muscle [16] has the fastest speed among the methods, but it has a high memory requirement and is not suitable for long-length sequences dataset. Therefore, current MSA methods are not applicable to some situations, such as calculating the conserved sequences from long-length sequences and large-scale datasets. With the aim of application to such datasets, in this work, we introduce a new method for conserved sequences based on a breadth-first search. Focusing on conserved regions instead of processing full-length alignment can greatly reduce the computational resource. Moreover, we applied our method to an influenza A virus HA dataset. Based on the conserved sequence, we designed the primers with high specificity and sensitivity for influenza A virus HA subtyping that were successfully tested via PCR experiments.

## 2. Materials and Methods

### 2.1. Influenza Virus Dataset

The sequence data used in this paper, including nucleotide sequence data and protein sequence data, are from the NCBI influenza database [17]. We selected sequence types “Protein” and “Nucleotide”; defined the search set as type “A”, the host as “any”, the country/region as “any”, and the protein as “HA”; selected subtype from H1 to H9 with N in “any”; and set the minimum sequence length to 560/1680 for the protein/nucleotide sequence and the collection date from 1918 to 2018. The sample numbers of each subtype are show in Table 1.

### 2.2. Conserved Sequence Searching

The software used to implement the algorithm is MATLAB (R2020a for windows, MathWorks, Inc., Natick, MA, USA).

In this research, we took amino acid sequences in place of nucleotide conserved sequences to process conserved sequence searching, as the replacement can reduce the sequence length by two-thirds with the similar information content, which can greatly improve the calculation efficiency.

Our algorithm of conserved sequence searching is based on a breadth-first search, adding a new amino acid at the end of the current conserved sequences to generate new candidate conserved sequences and selecting these sequences by recalculating their conserved probability in the global dataset.

#### 2.2.1. Protein Conserved Sequence

The protein sequence dataset was downloaded directly from the NCBI database (https://www.ncbi.nlm.nih.gov/genomes/FLU/Database/nph-select.cgi?go=database (accessed on 29 June 2021)). In other situations where only the nucleotide sequence dataset is available in the database, such as in the case of SARS-CoV-2 [18], amino acid sequences can be translated from the nucleotide sequence. Twenty amino acids were defined as length 1 conserved sequence string seeds. These seeds were the first 20 strings of queue ***q*** and their conservative rate is 100%. ***Q_i_*** is the ***i***th string in the queue, and ***d_j_*** is the ***j***th sample sequence of the current target subtype; ***q_i_*** and ***d_j_*** match when ***q_i_*** is ***d_j_***’s substring and ***f*** (***q_i_***, ***d_j_***) = 1. For the current string ***q_i_*** in the queue, we used Equation (1) to calculate ***q_i_***’s conservative probability.
(1)$$p_{i}=\frac{\sum_{j=1}^{n}f\left(q_{i},d_{j}\right)}{n},\;f\left(q_{i},d_{j}\right)=\{\begin{array}{c} 1\;if\;q_{i}\;and\;d_{j}\;match\\ 0\;otherwise \end{array}$$

When ***p_i_*** > 99% (the threshold set in this study is 99%), we consider ***q_i_*** as a conserved sequence of the target subtype; a new amino acid character is added to the end of the string to extend ***q_i_*** and new strings are added to the end of the queue.

#### 2.2.2. Nucleotide Conserved Sequence

We obtained the nucleotide conserved sequence candidates by locating the protein conserved sequences in the corresponding positions of nucleotide sequences. We used the “multialign” function in MATLAB to calculate the SNPs. The sequences with too many SNPs were not considered as nucleotide conserved sequences.

### 2.3. Polymerase Chain Reaction

PCR experiments were used to test our conserved sequence-based primer functions of clinical HA testing and subtyping. Influenza viruses are RNA viruses. In this study, we used their cDNA as substitute templates.

#### 2.3.1. Influenza Virus Template Plasmid

The template concentration was diluted to 100 ug/mL [19]. The plasmid (Sino Biological, China) information is shown in Table 2.

#### 2.3.2. PCR Primers Design

PCR primers were screened from nucleotide conserved sequences. We selected the primer pairs with similar annealing temperatures (△tm < 1 ℃). SNPs are unavoidable; therefore, we used degenerate bases (*n* < 5). Other principles follow the general requirements of primer design. The results are shown in Table 3.

#### 2.3.3. PCR Experiment

We used E. coli DH5α Competent Cells (Takara, Kusatsu, Japan, 9057) for the plasmid resuspension to culture bacteria and SanPrep Column Plasmid Mini-Preps Kit (Sangon, Shanghai, China, B518191) to extract plasmid. The cDNA of HA segments was separated by restriction enzyme (Takara, Kusatsu, Japan) digestion. Finally, we used a NANODROP 2000 (Thermo Scientific, Waltham, MA, USA) to measure the cDNA concentration and diluted it to 50 ng/μL. The PCR was performed in a 50-microliter system with 5 μL of 10X PCR buffer (Takara, Kusatsu, Japan, R001B), 0.25 μL of TaKaRa Taq (Takara, R001B), 1 μL of dNTP mixture (Takara, Kusatsu, Japan, R001B), 0.2 μL of forward primer (Sangon, Shanghai, China) and 0.2 μL of reverse primer (Sangon, Shanghai, China), 0.1 μL of cDNA, and 43 μL of nuclease-free water. Each primer pair tested 9 subtypes in the same condition and same batch. The PCR reaction conditions are shown in Table 4. Agarose gel electrophoresis images were collected using a gel imager (LIUYI, WD-9413B).

## 3. Results

### 3.1. Conserved Sequences

The protein conserved sequences (length ≥ 5) were mapped to the spatial structure of the influenza virus HA protein, and the visualization results are shown in Figure 1. Conserved sequences with conservative rates are listed in Appendix A.

As the results show, in H1–H9 subtypes, the conserved regions of the influenza virus HA protein are concentrated in the HA stem, while the HA head is less conserved. This is in accordance with the fact that virus antigen drift is more likely to happen on the antigen binding site. 

The protein conserved sequences were separately calculated from each subtype of the HA protein datasets. Focusing on subtype identification, only the conserved sequences with unique specificity for a single subtype were considered in the primer design. Figure 2 shows the most matching bases of nucleotide conserved sequences in different subtypes.

The conserved sequences are cut to a proper length to fit the PCR primer design. Figure 3 shows the designed primers’ matching in H1–H9 subtypes.

As Figure 3 shows, the conserved sequence-based PCR primers are expected to have a good specificity and sensitivity performance in HA subtype identification and are robust in global influenza virus strains.

### 3.2. PCR Experiment

We proved the feasibility of nucleotide conserved sequences in influenza virus hemagglutinin subtype identification via PCR experiments. The primers were selected from a calculated nucleotide conserved sequence set and were designed following the general primer design rules. To improve the sensitivity of primer pairs in matching the target subtype, we used degenerate bases to smooth the influence of SNPs (Figure 4).

The results regarding the length of the PCR amplification bands met our expectations. The nine primer pairs all have good performance in H1–H9 subtype identification with no PCR amplification bands on non-target subtypes lanes.

## 4. Discussion

The current MSA methods have bad performance in aligning datasets with long sequences and large sample sizes. Full-length MSA is unnecessary in many situations, such as conserved sequence analysis and primer design. In this study, based on the breadth-first search strategy, the time cost of our conserved sequence searching method depends on the conservative probability threshold setting and dataset sequence similarity. Taking the dataset containing m conserved protein sequences with L amino acids as an example, the time cost T∝(m×L×N). In addition, our conserved sequence results can be taken as anchors [29] to separate the long sequences into several shorter segments. Traditional MSA methods can be effectively optimized via the conserved sequence searching algorithm.

In clinical practicing, seasonal influenza testing relies on several early sample sequences. Focusing on the new mutations of the current strain, the designed seasonal influenza testing primers are unavailable in long-term testing of multi-strains [30]. It is also undeniable that focusing on the long-term effectiveness of testing primers negatively affects timeliness. The influenza primers designed based on conserved sequences cannot distinguish between epidemic and seasonal influenza strains.

Unlike testing primers, which are limited to a single strain, the ideal vaccines are expected to help vaccinated people to generate immunity against various seasonal strains. Using the conserved domain to induce immunity is a good choice. Recent research reported on a universal vaccine [31] replacing the HA head with the HA stem as the target domain to reduce the negative effect of antigenic drift on the long-term effect of the vaccine. However, as our results show, many mutations exist on the HA stem, and missense mutations can even change the protein structure, causing the antibody’s specific binding to fail. Selecting highly conserved sequences as a target domain is significant in designing a vaccine with long-term effectiveness.

Considering the length of the high conserved sequences, mRNA vaccines [32,33] and peptide vaccines [34,35] have potential in universal influenza vaccine research. Existing mRNA vaccine research [32] uses conserved sequences from multiple segments, including the HA stem, NA, M2, and NP, to strengthen the vaccine’s effect on the influenza virus antigen. Similarly, it is possible to design an mRNA vaccine using conserved sequences from different HA subtypes to provide broad cross-protection. Including the sequences from the HA stem [35], multitargeting is also feasible in peptide vaccine [34] design.

In general, our conserved sequence searching method displays good performance in a large-scale dataset. Our results regarding conserved nucleotide sequences and amino acid sequences are not only promising in influenza testing and HA subtype identification but also have high potential in future influenza research. Moreover, NA, another important antigen on the influenza virus surface with multiple subtypes, is suitable with the same methods and procedure as those for HA.

## 5. Conclusions

This study applied a conserved sequence searching algorithm based on a breadth-first search to an influenza A HA segment dataset and provided candidate sequences for long-term, effective testing primers and vaccine design for different HA subtypes. Via the PCR experiment, we proved the feasibility of conserved sequence-based primers in long-term influenza A virus HA testing and subtyping.

## Figures and Tables

**Figure 1 diagnostics-11-01328-f001:**
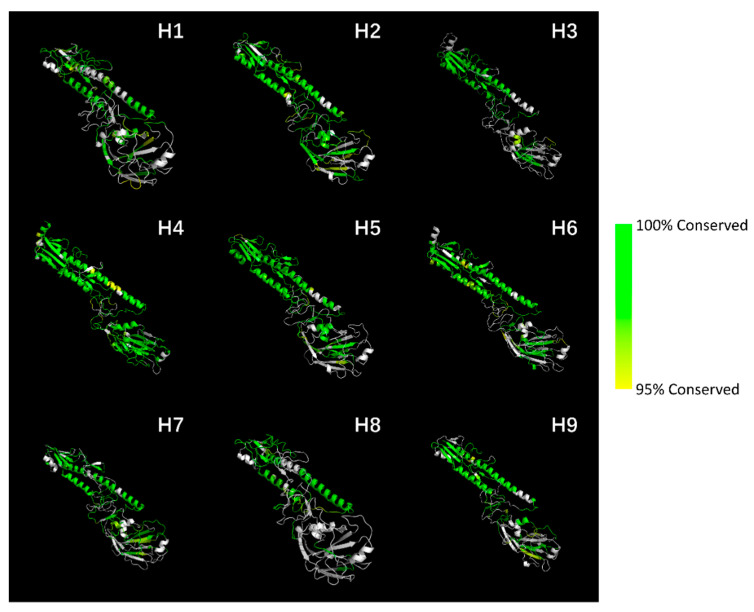
Visualization of HA protein conserved sequence mapping on spatial structure. The influenza virus HA protein is a homotrimer. We removed duplicate parts for enhanced presentation. The structures of H1 (1ruz) [20], H2 (2wr0) [21], H3 (4uo0) [22], H4 (5xl1) [23], H5 (1jsm) [24], H6 (5t08) [25], H7 (1ti8) [26], and H9 (1jsd) [24] are from the Protein Data Bank database [27]. The structure of the H8 subtype is from H1 structure’s homologous modelling [28]. The colors from green (RGB:010) to yellow (RGB:110) represent different conservative rates from 100 to 95%; conservative rates lower than 95% are colored in white.

**Figure 2 diagnostics-11-01328-f002:**
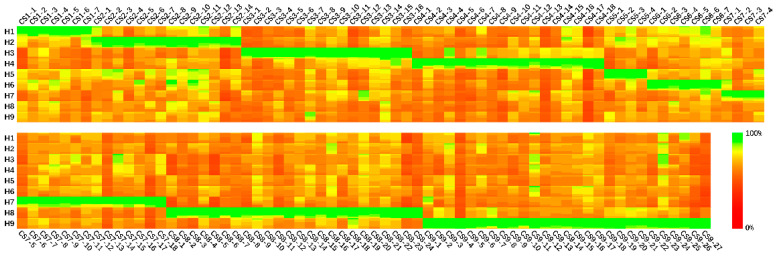
Nucleotide conserved sequence matching in H1–H9 subtypes. The matching of each nucleotide conserved sequence with each subtype is shown in 100 × 100 pixel squares, and each pixel represents an equal proportion of samples. Base matching of 100% is expressed in green (RGB010) and 0% base matching in red (RGB100). Row names are labelled by HA subtype, and column names are labelled by conserved sequences name: CS (conserved sequence) + HA subtype + order.

**Figure 3 diagnostics-11-01328-f003:**
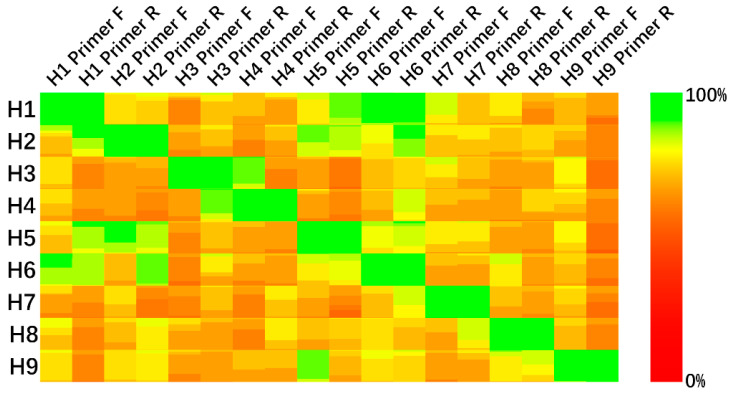
Matching of designed primers in H1–H9 subtypes. Each square contains 100 × 100 pixels, and each pixel inside the squares represents an equal proportion of samples. The HA subtype is labelled on the left of the figure; 100% base matching is expressed in green (RGB010); and 0% base matching is expressed in red (RGB100). Row names are labelled by HA subtype and column names are labelled by primer names.

**Figure 4 diagnostics-11-01328-f004:**
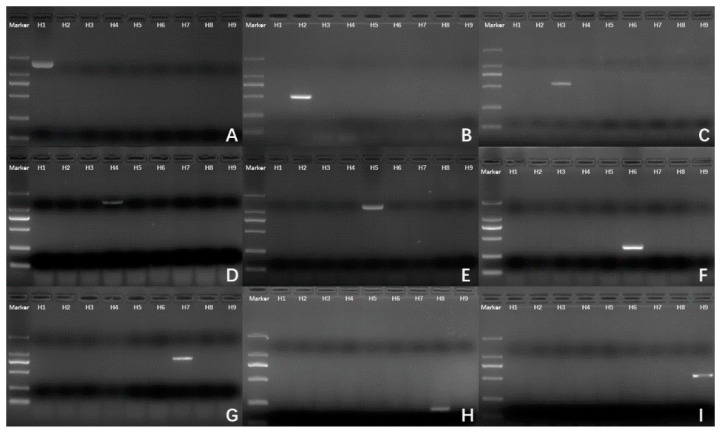
Agarose gel electrophoresis images of PCR experiment. (**A**–**I**) Using H1–H9-specific primers for H1–H9 cDNA (lanes marked as labelled). The marker used to the left side of each gel is a DNA ladder with the bands of 100, 250, 500, 750, 1000, and 2000 bp; the sample lanes are labelled from H1 to H9 in each gel. The original figures can be found in the Appendix A.

**Table 1 diagnostics-11-01328-t001:** Data size of the dataset.

Subtype	Nucleotide	Protein
H1	23,543	19,916
H2	613	624
H3	19,358	21,743
H4	1868	1896
H5	5658	6325
H6	1745	1788
H7	2090	2203
H8	139	141
H9	3623	3718

**Table 2 diagnostics-11-01328-t002:** Influenza virus template plasmid.

Subtype	Description	Catalog Number
H1	H1N1 (A/Beijing/262/1995) Hemagglutinin	VG11068-UT
H2	H2N2 (A/Guiyang/1/1957) Hemagglutinin	VG40119-UT
H3	H3N2 (A/Hong Kong/1/1968) Hemagglutinin	VG40116-UT
H4	H4N6 (A/Swine/Ontario/01911-1/99) Hemagglutinin	VG11706-UT
H5	H5N1 (Anhui/1/2005) Hemagglutinin	VG11048-UT
H6	H6N2 (A/chicken/Guangdong/C273/2011) Hemagglutinin	VG40398-UT
H7	H7N9 (A/Hangzhou/1/2013) Hemagglutinin	VG40105-UT
H8	H8N4 (A/pintail duck/Alberta/114/1979) Hemagglutinin	VG11722-UT
H9	H9N2 (A/Chicken/Hong Kong/G9/97) Hemagglutinin	VG40036-UT

**Table 3 diagnostics-11-01328-t003:** Primer design results.

Subtype	Protein Sequence	Primer Sequence	Fragment Length(bp)
**H1**	NVTVTHS	(5′–3′)AATGTRACWGTRACMCACTCW	1534
	SFWMCSN	(3′–5′)ATTRGARCACATCCARAARCT	
**H2**	YHHSNDQ	(5′–3′)TAYCAYCACAGCAATGAYCAR	481
	YQILAIYAT	(3′–5′)TGTAGCRTADATDGCAAGDATTTGRTA	
**H3**	ITPNGSI	(5′–3′)ATYACTCCAAATGGAAGCATY	532
	AEDMGN	(3′–5′)ATTKCCCATRTCYTCAGC	
**H4**	CYPFDV	(5′–3′)TGYTAYCCATTTGATGTG	1243
	QGYKDI	(3′–5′)RATGTCYTTGTATCCYTG	
**H5**	VTVTHA	(5′–3′)GTBACKGTYACACAYGCY	1219
	LMENERTLD	(3′–5′)RTCYAGAGTTCTYTCATTTTCCATGAG	
**H6**	WYGYHHE	(5′–3′)TGGTAYGGMTAYCAYCATGAR	349
	CFEFWHKC	(3′–5′)RCAYTTRTGCCARAATTCAAARCA	
**H7**	FYAEMK	(5′–3′)TTCTATGCRGARATGAAR	790
	GNVINW	(3′–5′)CCARTTWATSACATTVCC	
**H8**	EGMCYP	(5′–3′)GAGGGRATGTGYTAYCCT	175
	SINWLTKK	(3′–5′)CTTYTTRGTYARCCARTTRATGCT	
**H9**	GWYGFQHS	(5′–3′)GGTTGGTATGGDTTCCAGCATTCA	556
	AFLFWAM	(3′–5′)CATGGCCCAGAAYARGAAGGC	

**Table 4 diagnostics-11-01328-t004:** PCR reaction conditions.

Subtype	Step 1	Step 2			Step 3	Cycles
		S1	S2	S3		
H1	94 °C 5 min	94 °C 30 s	56 °C/2 min	72 °C/90 s	72 °C 7 min	30
H2			58.5 °C/1 min	72 °C/70 s		
H3			59 °C/1 min	72 °C/30 s		
H4			58.5 °C/2 min	72 °C/70 s		
H5			59 °C/2 min	72 °C/70 s		
H6			58 °C/30 s	72 °C/30 s		
H7			50 °C/1 min	72 °C/30 s		
H8			59 °C/30 s	72 °C/70 s		
H9			65 °C/1 min	72 °C/30 s		

## Data Availability

All sequence data are from the NCBI influenza database: https://www.ncbi.nlm.nih.gov/genomes/FLU/Database/nph-select.cgi?go=database (accessed on 29 June 2021).

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
