# Peer review of "Conserved Sequence Analysis of Influenza A Virus HA Segment and Its Application in Rapid Typing"

_diagnostics, 2021, doi:10.3390/diagnostics11081328_

Round 1

Reviewer 1 Report

The aim of the work “Conserved sequence analysis of influenza A virus HA segment and its application in rapid typing” was the selection of primers for the differential detection of influenza A viruses of different subtypes. A method for analyzing hemagglutinin sequences was developed to identify areas common to a subtype, but different in different subtypes. For each of subtypes 1-9, two such regions were selected, for which forward and reverse primers were synthesized. Plasmids with hemagglutinins of representative viral strains of these subtypes were obtained and the corresponding cDNAs were used as substitute templates.

It was shown that the selected primer pairs all have good performance in H1-H9 subtypes identification and no PCR amplification bands on non-target subtypes lanes.

The text in lines 126-128 and 130-131 got into the article by mistake

Reviewer 2 Report

Estimated Authors,

I've read with great interest your paper entitled "Conserved sequence analysis of influenza A virus HA segment and its application in rapid typing". Your research paper reports on a substantially novel feature from a classical topic on the virological research (i.e. genotyping of influenza Virus).

The present article may be both consistent with the aims of Diagnostics and deserving the actual publication because of its internal quality, but several improvements are, in my opinion, still required.

Firstly:

in the introduction, Authors report on several MSA algorithms: it is probably my fault, but I was substantially unable to understand how the Authors have innovated or employed the available instruments. In other words, How have the authors performed their MSA in different terms compared to the ref. [14-16], and how were the performances innovated? This should be discussed.

Second: Figure 1 is absolutely valuable in terms of design and accuracy, but it fails, at leat in my opinion, to report the conservative rates. I would suggest to implement some more schematic reporting, i.e. a table with the conservative rates of the assessed sequences.

Third: please perform a double-check of your text. Some typos are still scattered across the text (e.g. row 129-130).
